# Generating multivariate time series with COmmon Source CoordInated GAN (COSCI-GAN)

**Ali Seyfi**
Department of Computer Science
University of British Columbia
Vancouver, BC
aliseyfi@cs.ubc.ca

**Jean-Francois Rajotte**
Data Science Institute
University of British Columbia
Vancouver, BC
jfraj@mail.ubc.ca

**Raymond T. Ng**
Department of Computer Science
University of British Columbia
Vancouver, BC
rng@cs.ubc.ca

## Abstract

Generating multivariate time series is a promising approach for sharing sensitive data in many medical, financial, and IoT applications. A common type of multivariate time series originates from a single source such as the biometric measurements from a medical patient. This leads to complex dynamical patterns between individual time series that are hard to learn by typical generation models such as GANs. There is valuable information in those patterns that machine learning models can use to better classify, predict or perform other downstream tasks. We propose a novel framework that takes time series' common origin into account and favors channel/feature relationships preservation. The two key points of our method are: 1) the individual time series are generated from a common point in latent space and 2) a central discriminator favors the preservation of inter-channel/feature dynamics. We demonstrate empirically that our method helps preserve channel/feature correlations and that our synthetic data performs very well in downstream tasks with medical and financial data.

## 1 Introduction

Multivariate Time Series (MTS) are composed of individual time series (TS) sharing the same time reference. In some cases, the individual time series further share a common source such as the biometric values from a medical patient, the stock prices from economic events or geographically separated seismic measurements from a single earthquake. This leads to specific correlation patterns and time dynamics across the time series. Such complex patterns can be crucial when a model is trained on MTS and might need a huge amount of training samples to be captured by a machine learning algorithm. For many applications, however, there may not be enough high-quality training data. For example, for many biomedical and health care applications, data scarcity is common and data sharing to build a larger training set is challenging due to regulatory requirements or ethical concerns ([1], [2]). Such concerns are justified; it is well known that sharing data associated with a single individual, even anonymized, can lead to unexpected privacy breaches ([3], [4], [5], and [6]). Synthetic MTS could be an attractive alternative to share the patterns and statistical information of an MTS dataset. If done properly, synthetic MTS should not have a one-to-one mapping to the original data, although it can come with its own privacy and quality challenges (see [7] for example).

36th Conference on Neural Information Processing Systems (NeurIPS 2022).

Beyond data sharing, synthetic MTS can be used for augmenting a dataset to improve the performance of a trained model [8] or increase the contribution of underrepresented sub-populations [9]. For example, in the health domain, a patient can generate multiple TS from biometric measurements, wearable and IoT sensors, but the collection of such data may not have a representative coverage of a full population. Synthetic MTS can help to augment datasets for improving downstream analysis, such as for forecasting and classification tasks. In this work, we are interested in generating synthetic MTS that both preserve utility and statistical properties of the original data. By utility we mean the ability to support a specific downstream task such as the performance of a classifier when trained on synthetic data. Preserving statistical properties such as channel or feature[1] correlations increase the potential benefits to an unforeseen downstream task, exploratory data analysis or educational purposes when sharing data is not possible. We observe that current methods often struggle at correlation preservation needed for downstream task and our method addresses these limitations.

In this paper, we propose a novel method for generating MTS that comes from a common source by defining an architecture that explicitly takes inter-channel correlations into account. While MTS can be created with a typical generative deep learning architecture (such as VAE or GAN) with multiple channels output, we will demonstrate that our method not only preserves the quality of each individual TS by generating each individual TS separately from a common noise but also preserves the relationships between TS by having a central discriminator receiving all the generated individual TS as a single input. We consider *noise* to be a point of random sampling in the latent space that represents a patient's "whole biological environment." By common noise, we refer to a common point in the latent space, i.e., that latent space is the patient space, and we sample a patient by sampling a noise for the generator input.

We perform extensive empirical evaluation on MTS datasets where individual TS originate from a single source. We evaluate the resulting synthetic MTS by comparing their statistical features with the real MTS and their utility on downstream classification task. We pick classification to measure the utility of the synthetic data because classification is one of the most popular analytic tasks in machine learning. We also compare the real and synthetic MTS visually in embedding spaces, and evaluate our method against state-of-the-art baseline methods. Our contributions can be summarized as follows:

- To our knowledge, this is the first study to analyse how to generate multivariate time series with individual channel generation originating from a common noise while inter-channel correlations preservation is forced with a central discriminator.

- Demonstrating that COSCI-GAN compares favourably with state-of-the-art algorithms in downstream tasks on an Electroencephalography (EEG) eye state time series dataset.

- Demonstrating that COSCI-GAN results compares favourably with state-of-the-art algorithms in preservation statistical properties of on the EEG dataset.

- Open sourcing the implementation of our methods and experiments[2].

The rest of the paper is arranged as follows: Section 2 discusses related work. Section 3 formalizes he problem description. Section 4 presents our COSCI-GAN model architecture and gives implementation details. Section 5 presents an extensive empirical evaluation. Section 6 discusses the benefits, limitations and potential negative societal impacts.

## 2 Related Work

Synthetic data is often proposed to provide or to increase privacy protection [10]. The only privacy protection framework with predictable effect on privacy is to incorporate *differential privacy (DP)* into the learning algorithm ([11]) by adding calibrated noise to the parameter updates. For every non-DP generation method, the privacy must be assessed empirically with privacy attacks ([12]).

---

[1]Note that we use channel and feature interchangeably here since the term channel is commonly used for EEG and other medical time series that partly motivate this work. From a typical machine learning context, a channel is basically a feature. Generally in this paper, feature and channel corresponds to a collection of recorded data point from a given sensor.

[2]`https://github.com/aliseyfi75/COSCI-GAN`

Generative Adversarial Networks [13] are a popular method to generate synthetic data. Since their inception, they have expanded to include time-series data production, see [14] for a comprehensive overview. Mogren introduced the C-RNN-GAN approach, which employs RNNs as both the generator and discriminator in order to synthesise time series from a random vector [15]. Esteban et al. later proposed RCGAN to create medical data using a similar architecture [16]. These frameworks have been applied to a wide range of application domains, including biosignals [17], finance [18], sensor [19], text [20], and smart grid data [21]. However, the typical framework and loss function of GANs are insufficient for the production of multivariate time series, especially if we want to preserve the correlation among the channels as we will demonstrate below. Xu et al. developed COT-GAN based on ideas of optimal transport theory [22] and more recently, Li et al. developed TTS-GAN which is capable of generating realistic synthetic time series of any length [23] both work focusing solely on statistical evaluation of the data such as correlations.

Yoon et al. developed TimeGANs based on recurrent conditional GAN for capturing the temporal dynamics of data throughout time [24]. It entails training supervised and unsupervised targets concurrently using a learnt embedding space. It creates time series data by learning an embedding space and optimising it via binary adversarial feedback and stepwise supervised loss. Most recently, Fourier Flows [25] was proposed as a method based on a Fourier transform layer followed by a chain of spectral filters leading to an exact likelihood optimization.

## 3  Problem Formulation

Let $X$ be an MTS dataset of $N$ instances, each composed of $C$ channels $X_i$, where $i \in 1, \ldots, C$. An instance of $X$ can be described as $x^n = \{(x_1^n, \ldots, x_C^n)\}$. We want to find a distribution $q(X_1, \ldots, X_C)$ that is as close to $p(X_1, \ldots, X_C)$, the real distribution of our dataset, as possible. In the typical GAN framework, it may be difficult to find the best optimization solution for such a complex goal, which depends on the number of channels, duration, and distribution of the data. This is why we use separate generators $G_i$ to learn the marginal distribution of each channel, $p(X_i)$, separately, and then use a central discriminator to force preserving the real correlation between the channels by focusing on the conditional distributions $p(X_i|X_{i \neq j})$, where $X_{i \neq j}$ refers to all the channels excluding channel $i$. Figure 1 depicts this architecture, with more details discussed in the next section. Essentially, we have two objectives: a local and a central one.

**Local objective:**  In the local objective, the goal is to estimate the marginal distribution of each channel, $p(X_i)$; which means for each channel $i$, we should optimize:

$$\min_q D(p(X_i)||q(X_i)) \tag{1}$$

where D is any suitable measure of the distance between two distributions.

**Central objective:**  In the central objective, the goal is to estimate the conditional distribution of a channel given all the other ones, $p(X_i|X_{i \neq j})$. In order to illustrate this objective, we demonstrate it in a special case where the channels are independent of one another, which means we will have:

$$\min_q D(\prod_{i=1}^{C} p(X_i|X_{i \neq j})|| \prod_{i=1}^{C} q(X_i|X_{i \neq j})) \tag{2}$$

Our approach is that each generator $G_i$ share the same initial (noise) source $z$, such that Equation 2 becomes

$$\min_q D(\prod_{i=1}^{C} p(X_i|z)|| \prod_{i=1}^{C} q(X_i|z)) \tag{3}$$

We define the global loss to be a linear combination of the loss of local objective and loss of central objective.

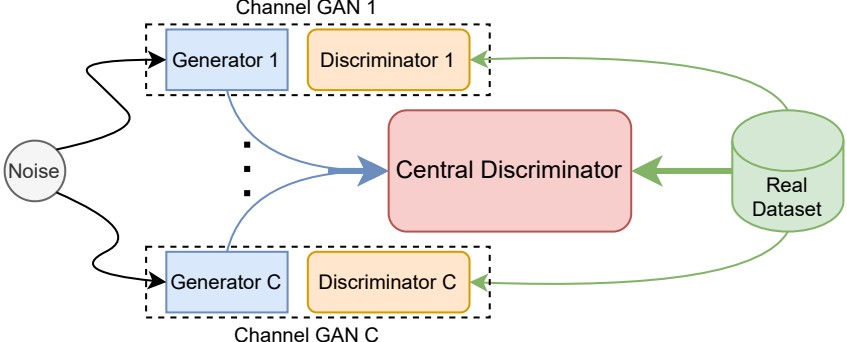

Figure 1: Overall structure of COSCI-GAN. Each Channel GAN is dedicated to one channel/feature with input either a real one or a fake one with their discriminator being a binary classifier. The central discriminator, a binary classifier, receives full instances (i.e. all channel) either all real or all fake.

## 4 COSCI-GAN

As shown in Figure 1, COSCI-GAN is made up of two main parts: 1) *Channel GANs*, which contain pairs of generator-discriminator dedicated to a single channel (univariate TS), and 2) the *Central Discriminator*, dedicated to all channels at once. Each of these parts is responsible for a specific task. In channel GANs, the generators are responsible for producing realistic TS and the discriminators are responsible for distinguishing between real and synthetic TS. The central discriminator is responsible for enforcing that all the generated TS of a given instance have the same correlation as those from real MTS. Producing all channels simultaneously necessitates that the generative model learn the joint distribution of all TSs, a difficult task that requires a substantial amount of data and time. In contrast, learning the marginal distribution of a single channel is a significantly simpler task. Consequently, the primary purpose of employing channel GANs as opposed to a single multichannel generator-discriminator pair is to assist each individual TS generator in synthesizing more accurate TSs from its own channel's data distribution. By including the central discriminator, we aim to enforce realistic correlations between the channels as much as possible.

### 4.1 Algorithm

Let our multivariate time series have dimensions of $N * L * C$, where $N$ is the number of instances in the dataset, $L$ is the length of each time series, and $C$ is the number of channels. As shown in Figure 1, there are $C$ pairs of Generator-Discriminator, or channel GANs. All generators are fed a shared noise vector to begin the generation process. Each generator in a channel GAN will synthesize a TS, and both the generated TS and the corresponding channel of real TS will be passed to their paired discriminator, which determines whether the generated TS is from the same distribution as the real ones. A pseudo-code of the COSCI-GAN algorithm is provided in Algorithm 1.

---

**Algorithm 1** COSCI-GAN

---

**for** epoch in epochs **do**
    **for** batch in training set **do**
        Create a noise vector, $Z$.
        **for** i = 1 to $n_{channels}$ **do**
            Extract $signal_i$ from the batch.
            Generate Fake signals $generated_i$ from $Generator_i$
            Train $Discriminator_i$ by feeding $signal_i$ and $generated_i$.
        **end for**
        Train $Central\ Discriminator$ by feeding
            $((generated_1, ..., generated_{n_{channels}}), (signal_1, ..., signal_{n_{channels}}))$.
        **for** i = 1 to $n_{channels}$ **do**
            Train $Generator_i$ with $Loss_{D_i}$ and $Loss_{CD}$.
        **end for**
    **end for**
**end for**

---

As mentioned earlier, the central discriminator's role is to preserve the inter-channel correlations. The TS synthesized by all channel generators will be concatenated as an MTS and fed to the central discriminator, which aims to determine whether the MTS is real or fake. We hypothesize that this will penalize unrealistic (un)correlation patterns between channels, and we have provided some evidences in the Supplementary Materials that other State-Of-the-Art (SOTA) methods often exaggerate or even create unrealistic correlations when compared to real data correlation.

## 4.2 Training

During training, the discriminators in the channel GANs (which we will refer to as channel Discriminators from now on), their paired generators, and the Central Discriminator will engage in a three-player game. The three-player objective of a given channel GAN combined with the central discriminator is:

$$\min_{\theta_i} \max_{\phi_i} \max_{\alpha} V(G_{i,\theta_i}, D_{i,\phi_i}, CD_\alpha) = \mathbb{E}_{x \sim P_{data}} \left[ log(D_{i,\phi_i}(x_i)]) + \gamma \cdot log(CD_\alpha(x)) \right]$$
$$+ \mathbb{E}_{z \sim P_z} [log(1 - D_{i,\phi_i}(G_{i,\theta_i}(z))$$
$$+ \gamma \cdot log(1 - CD_\alpha(G_{i,\theta_i}(z), G_{j \neq i}(z))] \tag{4}$$

where $G_{i,\theta_i}$ is the $i$-th generator with parameters $\theta_i$, $D_{i,\phi_i}$ is the $i$-th channel discriminator with parameters $\phi_i$, $CD_\alpha$ is the central discriminator with the parameters $\alpha$, $P_{data}$ is the distribution of the real time series, $x_i$ is the $i$-th channel of time series $x$, $G_{j \neq i}$ are all the other generators with fix parameters for the optimization step of $G_{i,\theta_i}$, $\gamma$ is a hyper-parameter that control the trade-off between well-preserving the correlation among the channels versus generating better quality signal within each channel, and $z$ is the shared noise vector sampled from $P_z$ distribution. The objective for all channels is a $2C + 1$ player game adding the terms with subscript $i$ in Equation 4.

In each epoch, we divide the MTS training dataset into a number of batches. If the batch size is assumed to be m, then each batch contains $x^{(1)}, \ldots, x^{(m)} \sim D$. For each batch, we sampled as many noise vector $z \sim P_z$ as the batch size. All channels' generators synthesize signals, and a gradient ascent step on the discriminators' parameters $\phi_i$s are taken. Then we concatenate the synthetic signals of all channels and use them in addition to the real data to take a gradient ascent step on the central discriminator parameters $\alpha$. In the end, we take a gradient descent step on the generators parameters $\theta_i$s.

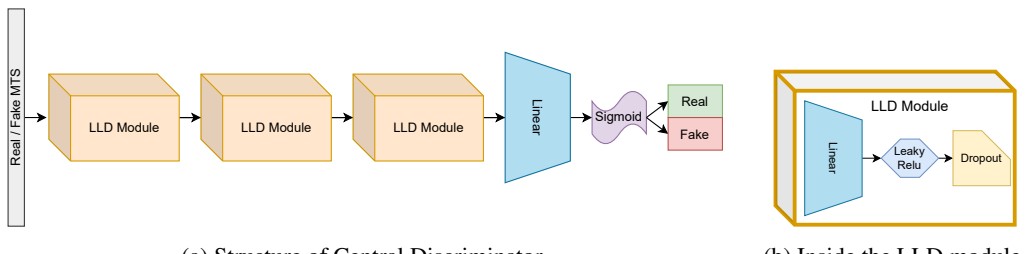

(a) Structure of Central Discriminator.       (b) Inside the LLD module.

Figure 2: Structure of an MLP-based Central Discriminator.

## 4.3 Key Implementation Details

There are many possible choices of network types for channel GANs. We show in the Supplementary Materials that networks based on Long short-term memory (LSTM) generate signals of higher quality than the networks based on Multi-layer perceptron (MLP).

We have investigated LSTM and MLP based networks for the central discriminator. Although LSTM-based networks perform better than MLP-based networks in Channel GANs, MLP-based networks perform better in the Central Discriminator. We hypothesize that if the central discriminator is too powerful, the results will be of lower quality, as the generators will strive to make the signals more correlated at the expense of realistic individual TS.

Figure 2a depicts the structure of an MLP-based central discriminator. It consists of three Linear-LeakyReLU-Dropout (LLD) modules, a linear module, and a sigmoid function. Figure 2b demonstrates the structure of an LLD module.

# 5 Empirical Evaluation

## 5.1 Toy Sine Datasets: Diversity vs Correlation Preservation

To be able to investigate the empirical behavior of COSCI-GAN and, in particular, the effect of the central discriminator, we need to have complete control over the nature of the datasets, particularly the ground truth of our desired tasks. Thus, we simulated three "toy medical" datasets with two channels and used them as "real" datasets to generate synthetic data. For all these datasets, we assumed that each instance correspond to a different patient, and each patient produce measurements for two channels ($c_1$ and $c_2$). To make the datasets a bit more realistic, we also assumed that there are two types of patients ($pt_1$ and $pt_2$), as in "healthy" vs "condition". The three datasets are:

- **Simple Sine** is derived from the formula: $x = A \sin(2\pi f t) + \epsilon$. The difference between the signals is that the amplitude ($A$) for patient type 1 comes from $\mathcal{N}(0.4,\, 0.05)$, whereas the amplitude for patient type 2 comes from $\mathcal{N}(0.6,\, 0.05)$. The other difference is that Channel 1 has a frequency ($f$) of 0.01, while channel 2 has a frequency of 0.005. In addition, the noise $\epsilon$ comes from $\mathcal{N}(0,\, 0.05)$.

- **Sine with frequency changes** contains the same signals as *Simple Sine*, except that the frequency of all sine functions doubles exactly in the middle of the time series. This allows us to examine the situation with varying frequencies.

- **Anomalies** is created by replacing the middle of the time series with Gaussian noise, thus allowing us to examine the impact of anomalies.

The visualization of all three toy datasets are available in the Supplementary Materials.

We assessed the behaviour of COSCI-GAN, particularly the central discriminator, by two criteria:

(1) Diversity: requiring that the generators should synthesize from both patients' distributions and there should be no *mode collapse*, a common failure of GANs, which occurs when the generator fails to produce results as diverse as the real data. We measured the diversity by comparing the distribution of amplitude of patients' signals in the real dataset, which is a bimodal Gaussian distribution, with the distributions of amplitude of the generated samples using Wasserstein Distance (WD). We took the WDs average (AWD) to aggregate across the channels. A lower AWD indicates a closer similarity to the real distributions. The first column of Table 1 shows the AWD for the various cases.

(2) Correlation Preservation: requiring that the amplitudes of channels for each patient types should be equal to each other as much as possible. We measured the amplitudes of channel 1 and 2 in all signals and verified their similarity. We defined our correlation metric as the average euclidean distance (AED) between the amplitude mapped on a 2D plane (Channel 1 vs Channel 2) and a line with slope 1. The resulting plot is provided in the Supplementary Materials, and the numeric values are summarized in Table 1. A lower AED indicates stronger preservation of correlation.

Table 1: Results of Diversity (AWD) V.S. Correlation Analysis (AED)

| Dataset | Method | AWD | AED |
|---|---|---|---|
| Simple Sine | Without CD | **0.0472** | 0.1326 |
| | With CD | 0.0800 | **0.0177** |
| Freq changes | Without CD | **0.0397** | 0.0769 |
| | With CD | 0.0679 | **0.0242** |
| Anomalies | Without CD | **0.0540** | 0.0766 |
| | With CD | 0.0726 | **0.0161** |

As clearly shown in Table 1, there is a trade-off between diversity and correlation preservation. That is when we used the CD, the correlation was better preserved, but at the expense of diversity (similarity between the generated time series' marginal distribution of amplitudes and the toy Simple Sine time series' marginal distribution of amplitudes). Conversely, not using the CD would allow the generated sample distributions to be closer to the real data, but the generated channels were less correlated. The strength of the CD is controlled by the parameter $\gamma$ as shown in Equation (4). As we conducted experiments to tune the hyper-parameter $\gamma$, we observed that there is a stable range for $\gamma$. We ended up setting $\gamma$ to 5, which provides stable results for all the experiments presented in this paper, whether the datasets are the toy medical ones or the real ones to be discussed later.

## 5.2 Toy Sine Datasets - Feature-based Correlation Analysis

The *catch22* feature set has been introduced to capture 22 CAnonical Time series CHaracteristics commonly seen in diverse time series data mining tasks [26]. Using this feature set, we assessed how correlation between any pair of catch 22 features was preserved in synthetic time series data generation. In other words, if a pair is strongly correlated in the real dataset between the two channels, we would like to see that preserved in the synthetic dataset. Similarly, if two features are not correlated in the real dataset, they should remain uncorrelated in the synthetic dataset.

Figure 3 shows three heatmaps for those pairwise correlations between the two channels of simple sine dataset (The same figure for other toy datasets are provided in the Supplementary Materials). To simplify the heatmaps, we removed 7 features that were constant among the real datasets, and kept the remaining 15 features that varied. The left heatmap shows the pairwise correlations of the 15 features for the real dataset. The centre heatmap is the one for the synthetic dataset generated by COSCI-GAN without a CD, whereas the right heatmap is the one generated by COSCI-GAN having the CD. Clearly, the right heatmap resembles the left heatmap much more closely. The centre heatmap shows that without the CD, almost all the correlation relationships of the 15 features were destroyed.

While the heatmaps are useful for visualization, we also compared quantitatively the correlation matrices between the two channels using various metrics: (1) Mean Absolute Error (MAE), (2) Frobenius norm, (3) Spearman's $\rho$, and (4) Kendall's $\tau$. For MAE and the Frobenius norm, a smaller value indicates greater similarity between the correlation matrices of the real and synthetic datasets. For Spearman's coefficient and Kendall's coefficient, the closer the value is to 1, the higher is the similarity. Results shown in Table 2 provide convincing evidence of the effectiveness of the CD in synthetic MTS generation.

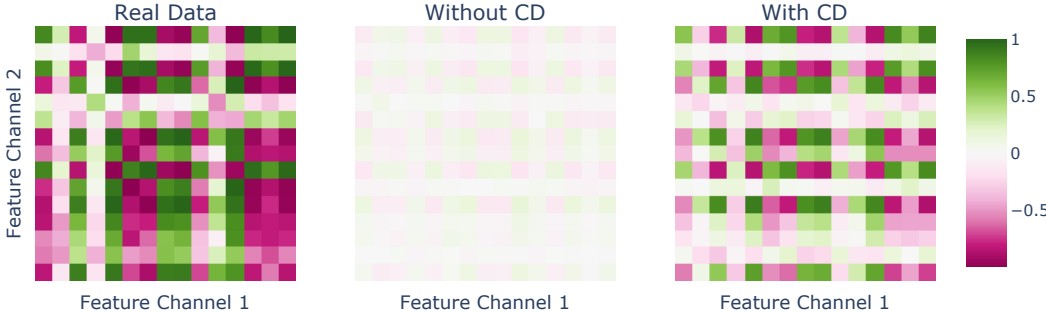

Figure 3: Heatmaps for Catch22 Pairwise Correlations.

Table 2: Similarity between Correlation Matrices

| Dataset | Method | MAE | Frobenius norm | Spearman's $\rho$ | Kendall's $\tau$ |
|---|---|---|---|---|---|
| Simple Sine | Without CD | 0.671 | 11.295 | -0.761 | -0.569 |
| | With CD | **0.298** | **5.666** | **0.848** | **0.700** |
| Freq changes | Without CD | 0.268 | 7.889 | 0.259 | 0.174 |
| | With CD | **0.131** | **3.413** | **0.834** | **0.661** |
| Anomalies | Without CD | 0.289 | 8.113 | 0.428 | 0.297 |
| | With CD | **0.199** | **5.362** | **0.786** | **0.612** |

## 5.3 EEG Eye State Dataset and Downstream Classification

We selected a 14-channel EEG eye state dataset to measure the effectiveness of COSCI-GAN on real signals [27][3]. This dataset contains a label indicating whether the patient's eyes were open or closed (1 indicates closed, and 0 indicates open). Each time series is 117 seconds long, resulting in 14980 samples at a sampling rate of 128 per second. To remove outliers from the dataset, we experimented with various z-score values and determined that a value of 3 is optimal for our dataset, so we removed points with z-scores greater than 3, the same value as other EEG studies (e.g. [28]). The label of the dataset allows us to create a downstream eye blink classification task as follows.

---

[3]https://archive.ics.uci.edu/ml/datasets/EEG+Eye+State with license details therein

We extracted a window of 800 samples in length containing an eye blink, with a margin of 200 samples at the beginning and end of each eye blink, and labelled those frames with 1. We also extracted 800 samples that did not contain an eye blink, with a margin of 200 samples between the beginning and end of an eye blink, and labelled them with 0. We now have 1024 frames of each label for classification. Then we performed a forward feature selection, and chose top 5 channels regarding the accuracy of our classification task.

To measure the effectiveness of COSCI-GAN for classification, we used the approach of train-on-real and test-on-fake, and the opposite approach of train-on-fake and test-on-real. We measured the accuracy of an LSTM-based classifier, which is described in details in the Supplementary Materials, on a dataset that contains two channels. Table 3 shows the results of this comparison. We compared the accuracy of the classifier when the synthetic data were generated with and without the CD. Once again, the CD brings significant value to downstream classification tasks.

Table 3: Accuracy in Classification Task

| Experiment | COSCI-GAN with CD | COSCI-GAN without CD |
|---|---|---|
| Train-on-real, Test-on-fake | 0.790 | 0.644 |
| Train-on-fake, Test-on-real | 0.634 | 0.561 |

Next we compared COSCI-GAN against a baseline method for generating MTS data. The baseline method is an LSTM-based GAN that simultaneously generated all channels. The same LSTM network was utilized in the baseline method and the COSCI-GAN modules. The only difference is that the output layer of networks in the baseline method must generate all channels rather than just one. Below we show the results of two experiments: (1) the All-synthetic experiment, and (2) the Augmentation experiment.

**(1) All-synthetic experiment**   In this experiment, we assessed how well COSCI-GAN performed in classification task when compared with the baseline method and the actual dataset. We performed cross-validation by using 80% of our real dataset for training the GANs. Then only the synthetic data were used to train the classifiers, which were then tested on the hold-out 20% of the real dataset. We investigate the utility of the synthetic data for different number of channels in Figure 4a. We repeated each experiment 30 times with different random seeds for each setting, and statistical significance tests were done on the boxplots. A further comparison including COSCI-GAN without the CD is provided in the Supplementary Materials.

Figure 4a demonstrates that as the number of channels is increased, the accuracy of a classifier trained on data generated by COSCI-GAN and evaluated on real datasets increases. In contrast, the baseline method went the opposite way and demonstrated that the performance of MTS generation with all channels together degrades as the number of channels increases. As the number of channels grew, synthetic data generated by COSCI-GAN gave a similar average performance as the real dataset.

**(2) Augmentation experiment**   This experiment was set up exactly like the previous one, but instead of using only synthetic time series to train the classifiers, we augmented the real dataset with an equal number of synthetic training samples. In Figure 4b, the boxplots for the real datasets across the different number of channels are exactly the same as those in Figure 4a for the real datasets.

Between Figure 4a and Figure 4b, COSCI-GAN shows significant improvements in accuracy. The difference was that the synthetic data generated were added to the real data. COSCI-GAN still outperformed the baseline method both in terms of the median and the variations in accuracy.

Figure 4b was based on an augmentation ratio of 1:1. Figure 4c shows a scatter plot of comparing the accuracy of COSCI-GAN (y-axis) against the baseline method (x-axis) across six different augmentation ratios: $1:1, 1:2, 1:4, 1:6, 1:8, 1:10$, i.e. augmenting with up to 10 times more synthetic data than real training data. We repeated the experiment five times with different random seeds for each of these settings; the accuracy of each run is plotted in Figure 4c. Thus, any point above the diagonal red line indicates that COSCI-GAN outperformed the baseline method. The points in the figure are colour-coded based on the number of channels. It is obvious that COSCI-GAN dominated the baseline method when there is more than two channels. To further quantify the differences in accuracy, we add dotted diagonal lines in 4c, representing a difference in accuracy with increments of 0.1. For instance, the lowest dotted diagonal line represents the cases when the accuracy of COSCI-GAN is 0.1 below that of the baseline method. Conversely, the other three dotted diagonal

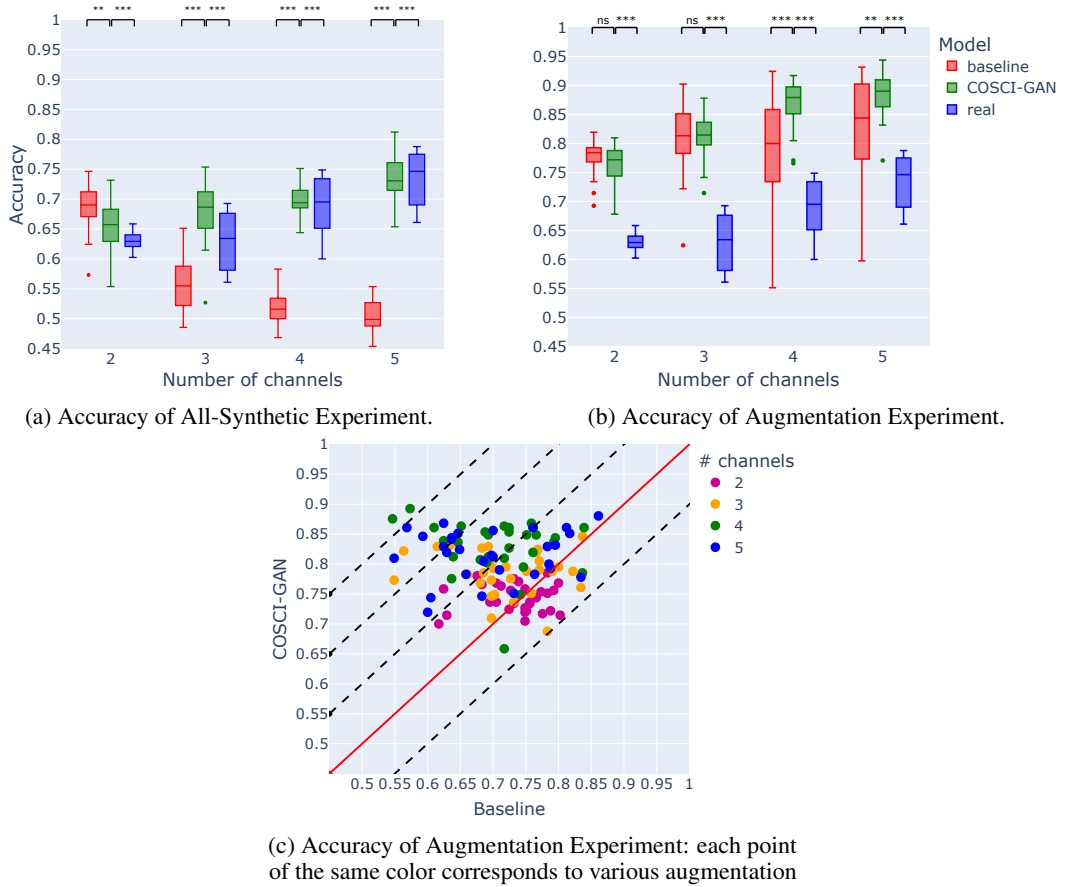

(a) Accuracy of All-Synthetic Experiment.
(b) Accuracy of Augmentation Experiment.

(c) Accuracy of Augmentation Experiment: each point of the same color corresponds to various augmentation ratios, from $1:1$ to $1:10$, see text for details.

Figure 4: Experiment three results. In figure (a) and (b), T-tests were applied to the results in order to show whether there is a statistically significant difference between the distribution of the results. **ns** means no significant difference, **\*\*** means $0.001 <$ p-value $< 0.01$, and **\*\*\*** means p-value $< 0.001$.

lines represent the situations when the accuracy of COSCI-GAN is 0.1, 0.2 or 0.3 better than that of the baseline. Figure 4c shows that COSCI-GAN almost always performed better than the baseline method, up to 0.3 higher accuracy, and most exceptions are from the 2 channel dataset where the accuracy is at most 0.1 lower.

### 5.4 Comparing with State-of-the-art methods on EEG Classification

In the final experiment, we compared COSCI-GAN with two state-of-the-art (SOTA) methods: TimeGAN[4] [24] and the most recent Fourier Flows [5] [25] discussed previously. In these papers, the downstream task was predicting the next time point of the time series; i.e., forecasting. Because our focus is on classification, we used TimeGAN and Fourier Flows' code to generate the five EEG channels that we chose in the previous experiment and performed the augmentation experiment that we described in 5.3. As shown in Figure 5 COSCI-GAN leads to the best classification accuracy and there is a statistically significant difference between the results of COSCI-GAN and two other methods.

To compare the statistical properties of COSCI-GAN with SOTA methods, we repeated the correlation analysis from 5.2. Figures similar to Figure 3 for TimeGAN and Fourier Flows are provided in the Supplementary Materials. We computed the MAE between the real correlation matrix and each method's correlation matrix for each pair of channels in the EEG dataset. Table 4 shows that COSCI-GAN, in addition to giving better classification accuracy, provides the closest similarity in inter-channel correlations to the real dataset.

---

[4]https://github.com/jsyoon0823/TimeGAN with license details therein
[5]https://github.com/ahmedmalaa/Fourier-flows with license details therein

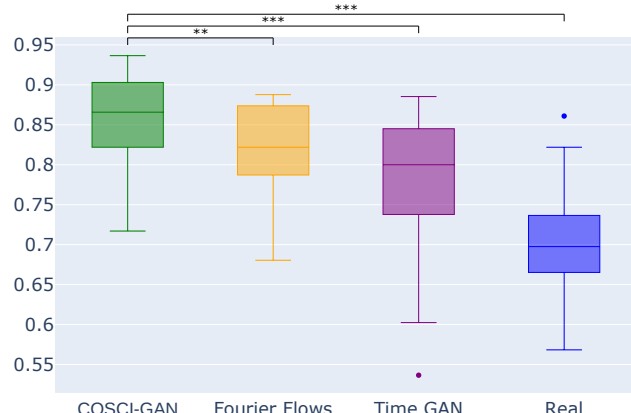

Table 4: Similarity between Correlation Matrices in EEG dataset mean ± standard deviation

| Method | MAE |
|---|---|
| COSCI-GAN | **0.111 ± 0.005** |
| TimeGAN | 0.257 ± 0.008 |
| Fourier Flows | 0.146 ± 0.006 |

Figure 5: Accuracy for the Augmentation Experiment Comparison with SOTA methods

In TimeGAN and Fourier Flows papers, a daily historical Google stocks dataset from 2004 to 2019 was used, and PCA[29] and t-SNE[30] plots were shown to compare their diversity in compare with the real data. We repeated this experiment using data from TimeGAN repository [6], and showed that COSCI-GAN samples were more diverse and distributed in a way that was more similar to the real dataset distribution. The figures of this experiment are provided in the Supplementary Materials.

## 6 Discussion

In this paper, we introduced COSCI-GAN, a novel framework for multivariate time-series generation that delivers more correlated channels. By preserving the correlation between channels, COSCI-GAN is able to generate time-series that are more similar to the real time-series and achieve better performance in downstream classification tasks than other state of the art methods.

We have shown that our framework is relevant for generating MTS from a common source and we argue that it is particularly suited for human-based biometric measurements. In our experiments, we have never had performance limitations, we foresee however, that COSCI-GAN will not scale to a very large number of channels as a dedicated GAN for each channel is needed. However, this issue is not exclusive to the COSCI-GAN method. It is a "computing resources" limitation for the COSCI-GAN method, whereas it is a fundamentally intractable problem for the baseline and many other methods. In addition, COSCI-GAN has the benefit of being parallelizable, which makes it faster. On the other extreme, we have shown that COSCI-GAN is not competitive for two channels where it often performed worse than a simple baseline.

It is worth re-iterating that synthetic data generation does *not* guarantee privacy and similarly, there is no way to know in advance the performance on a downstream task so both characteristics should be empirically evaluated post-generation. Outliers and minorities are often affected most by privacy leaks as they do not get protected by a large number of similar data samples. We also acknowledge that synthetic data generation can cause harm propagating and even magnifying bias from the data it is based on.

As future work, our method could be extended to more practical use cases where the various channels corresponds to different types of time series, e.g. heartbeats, temperature, respiration and wearable measurements and so on. As another extension, we could consider having an initial noise embedding (corresponding to several initial noises) that are all originated from a single source, in order to have more control on each channel's distribution. On the technical side, our framework can be implemented with a wide variety of GANs chosen based on the data type including modern architectures such as transformers. Additionally, as stated previously, Channel GANs could train in parallel, which would accelerate the training process. The code we have provided is modular and its parallelized version can be implemented in future.

---

[6]https://github.com/jsyoon0823/TimeGAN/blob/master/data/stock_data.csv

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
