# OpenReview forum: "Generating multivariate time series with COmmon Source CoordInated GAN (COSCI-GAN)"
_NeurIPS.cc/2022/Conference — NeurIPS 2022 Accept_

### Official Review · Reviewer_oPqB · 2022-07-09

**Rating:** 4
**Confidence:** 4
**Soundness:** 2 fair
**Presentation:** 3 good
**Contribution:** 2 fair

**Summary:**

This paper proposes a generative method for multivariate time series. The method consists of individual GANs for each channel of time series with a shared initial noise, and a global discriminator for preserving the correlations between different channels. The design is to model the common factors of different channels.

**Questions:**

1. In Eq 2, why to assume the conditional distributions are independent in the conditions (channels)?
2. Will the shared initial noise in Eq 3 and the modeling of the joint condition in Eq 2 have similar effects? How could they both be indispensable in the method?
3. Is it reasonable to add several initial noises, instead of one, for improving the flexibility?


**Limitations:**

The authors addressed the potential negative societal impact well.

**Strengths And Weaknesses:**

Strengths

1. The problem of synthetic time series generation is important in many tasks, e.g., data sharing, data augmentation.
2. The proposed method uses a reasonable approach to enforce common factors of multivariate time series and preserve correlations.
3. The experiments on synthetic and real datasets indicate the effectiveness of the proposed method to some extent.

Weakness:

1. The method is sort of incremental on GAN for time series generation. Preserving correlations using a central discriminator is similar to enforce a regularization of different channels (Fig 1), which is a common way of coordinating multivariate time series.
2. In Eq 2, the condition conditional distributions are assumed independent. That is, the equivalence between the joint condition and the product of individual condition. This is not clearly justified.
3. Enforcing a shared initial noise in Eq 3 and modeling the joint condition in Eq 2 seem have similar effects of regularizing different channels of time series. The description is not clear on how could these two parts both be important.
4. Enforcing a single initial noise may be a strong constraint. Adding K initial noises, where K is smaller than the number of channels, could make the model more flexible and fit training data better.
5. In the experiments, mode time generative model could be compared, on both synthetic and real datasets for comprehensiveness and consistency.

---

> ### Author Response · Authors · 2022-08-01
> **Answers to the questions**
>
> Regarding your first question, there is no such independence assumption in our algorithm; however, we simplified the equations with this assumption to help the reader understand the concept and the general idea. To avoid misunderstandings, we will clarify this point in the final release of the paper.
>
> In response to your second question, Equations 3 and 2 are actually equivalent. Equation 3 is the Group GAN implementation of Equation 2.
>
> Your last question is a great idea, though it should probably not be several "independent" noises at first. We could consider having an initial (patient) embedding (corresponding to several initial noises) in future work, but it would originate from a single source. We will mention it as a promising next step in future works.

---

### Official Review · Reviewer_zeED · 2022-07-11

**Rating:** 5
**Confidence:** 3
**Soundness:** 3 good
**Presentation:** 2 fair
**Contribution:** 2 fair

**Summary:**

The authors present GroupGAN for synthesis of multivariate time-series data.  In contrast to previous approaches where features are jointly learned, GroupGAN trains a separate GAN while also training a central discriminator which classifies feature vectors as either real or fake.  The intuition behind the latter is to preserve correlations among the individually trained generators.  Synthetically trained data are evaluated for low-dimensional synthetic time-series data and supervised EEG data.

**Questions:**

The following is a major contribution of the paper: "this is the first study to analyse how to generate multivariate time series
with individual channel generation originating from a common noise while inter-channel
correlation preservation is forced with a central discriminator." <- How realistic is this scenario?  E.g., for the listed examples "the biometric values from a medical patient, the stock prices from economic events or geographically separated seismic measurements from a single earthquake," do these assumptions hold?  E.g., while biometric values are collected from a single medical patient, different signals may have different noise properties (e.g., ECG, blood pressure, etc.).

We hypothesize that this will penalize unrealistic (un)correlation patterns between channels."  <- Can you supply evidence to support
 this, or point out where in the text evidence will be demonstrated?

Suggestions:
" it is well known that sharing data associated with 26 single individuals, even anonymized, can lead to unexpected privacy breaches" <- Well known ML examples:
          -Narayanan, Arvind, and Vitaly Shmatikov. "Robust
          de-anonymization of large sparse datasets." 2008 IEEE
          Symposium on Security and Privacy (sp 2008). IEEE, 2008.

          -Gong, Maoguo, et al. "A survey on differentially private
          machine learning." IEEE computational intelligence magazine
          15.2 (2020): 49-64.

**Limitations:**

A discussion of the tradeoff between learning a single multivariate GAN vs multiple GANs would help strengthen the paper.  While the authors have mentioned learning a large number of individual GANs can be prohibitive for high-dimensional data, in situations where jointly learning the distribution of all features is intractable, the presented architecture is highly parallelizable while maintaining a mechanism to encourage the separately trained GANs generate correlated data.

**Strengths And Weaknesses:**

Strengths: The proposed architecture is novel and interesting.  The central disciminator, which enforces preservation of correlations among independent GANs, is intuitive and could be helpful regularizing different GAN instances to promote consistency/correlations.  While the authors claim: "The main reason for having channel GANs as opposed to a single multichannel generator discriminator pair is to make each generator powerful in its own channel distribution, and by including the central discriminator, we enforce realistic correlations between the
channels as much as possible."
It seems like another major advantage of this approach is tractability; jointly modeling all TS at once may lead to a computational explosion, but the current setup allows the computationally efficient benefits of training individual generators while also ensuring correlations are learned among the individual TS through the central discriminator.

Weaknesses: The writing and description of the model could be much improved to align with time-series literature.  In particular, the description and focus on channels both confuses and limits the paper to vision data.  Channels typically refers to vision data, but moving the model description and text to describing time signals as comprised of features broadens the work and its applicability (i.e., a large number of time series work are not vision data).  Furthermore, the description in terms of channels breaks down for different applications (i.e., MFCCs in speech recognition).  The clearest/most general description in time-series analysis is a (possibly multirate as well as multivariate) process is composed of various features collected at different times.  Furthermore, there seems to be a general misconeption on what a time series is in the paper; the EGG eye state data is a single time series, each time-instance of which consists of 14 features (or measurements) and a binary label.

Evaluation of the proposed method requires significant improvement.  The synthetic data is extremely low-dimensional (only 2 dimensions) and resulting hyperparameter tuning, performance, an(d (particularly) catch22 results on this data are thus unconvincing.  Furthermore, the real data is preprocessed in an extremely ad-hoc way (lines 229-234) without any explanation.  E.g., the EEG data is supervised, why are labels re-estimated during preprocessing?  How many outliers were present in the data, and what observed effects on GroupGAN and competitors warranted removal?

Necessary details are also lacking from the paper.  In particular: "The baseline method is an LSTM-based GAN that generated all of the
channels simultaneously" <- this competitor requires much more explanation.  What were the exact details of the architecture and learning schedule?  Why were they chosen?  Why were existing methods (i.e., TimeGAN)  not compared to?

---

> ### Author Response · Authors · 2022-08-01
> **Answers to the questions**
>
> As you mentioned, the word “channel” could be misleading, and in the final release of the paper, we will clarify what we mean by a channel. The term "channel" was derived from the neuroscience literature on EEG signals, such as the papers "Imaging human EEG dynamics using independent component analysis" by Julie Onton et al., "Fundamental of EEG measurement" by M. Teplan, and many others.
>
> The goal of the low-dimensional experiments was to show that Group GAN does indeed do what our intuition suggests.
>
> In response to your concern about why we re-estimated the labels during the preprocessing phase, we did so because, in our experience, capturing an eye blink with EEG data is easier than stating the openness or closeness of the eye for a deep learning classifier. To achieve good classification results using real-world data, we decided to perform an eye blink detection task instead and re-estimate the labels.
>
> There are some outliers in the EEG time series due to measurement tool imperfections and noise. Filtering EEG signals is common in neuroscience tasks; a review of the possible methods for removing artifacts from EEG signals is provided in "Removal of Artifacts from EEG Signals: A Review" by Xiao Jiang et al.. In their paper "Faster: Fully Automated Statistical Thresholding for EEG Artifact Rejection," Nolan et al. used the z-score to detect outliers.
>
> The baseline method is the same LSTM-based network we use in GroupGAN, which takes a noise vector and generates all channels simultaneously. This information will be included in the final version of our paper. Because there was insufficient space in the main paper, the exact details of the architecture are provided in supplementary materials, and the implementation of all networks will be available in our GitHub repository to demonstrate the precise structure and parameter values. The choice of parameters is the results of various experiments to find those that produce good results on the validation dataset for both Group GAN and Baseline methods. In addition to the baseline method, we also compared our method to existing state-of-the-art methods such as TimeGAN in the last section.
>
> Regarding your first question, we need to clarify what we mean by “noise”. We consider "noise" to be a (random) sampling point in the latent space that represents one patient's "whole biological environment." By common noise, we actually mean a common point in the latent space, i.e., that latent space is the patient space, and we are sampling a patient by sampling a noise for the generator.
>
> Regarding your second question, we have demonstrated in Figure 9 of the supplementary material that other SOTA methods often exaggerate or even create unrealistic correlations when compared to real data correlation. In the final version of our paper, we will explain the reasoning behind the hypothesis if the paper is accepted.
>
> Thank you for the suggested papers. We will include the mentioned papers in the related works in the final release of our paper.
>
> You are completely correct about the limitation point. In the final version of the paper, we will discuss the tradeoff between the challenges of learning a single multivariate GAN versus multiple univariate GANs. As you mentioned, other methods will also perform poorly in high-dimensional data. In the GroupGAN method, it is a "computing resources" limitation, whereas it is an intractable problem for the baseline and many other methods, which is more fundamental. As you mentioned, GroupGAN has the advantage of being parallelizable, which allows it to be faster. These discussions will be included in the final version of the paper, and parallelization will be mentioned as a promising next step in future works. Thanks a lot for your constructive suggestions.

---

> > ### Comment · Reviewer_zeED · 2022-08-09
> > **Response to the authors**
> >
> > I thank the authors for their response.
> >
> > I agree, an early definition of what is meant by "channel" and "noise" would greatly help readers understand the paper.  As previously stated, an average ML reader will most likely understand "channel" to refer to rgb dimensions in vision data.  Thus, I strongly encourage the authors to avoid the general use of "channel" in favor of "feature."
> >
> > Preprocessing of raw data is understandable, but should also be agnostic to downstream methods being evaluated.  Most importantly, how was the outlier threshold (z-scores > 3) chosen?  For the statement: "Then we performed a forward feature selection, and chose top 5 channels regarding the accuracy of our classification task."  How was feature selection performed?  Why are only 5 channels chosen, and under what criteria was 5 selected?  As previously mentioned, thorough evaluation of GroupGAN is a concern.  How does the method fair for a standard ML dataset (e.g., much higher dimensionality than 5)?

---

> > > ### Author Response · Authors · 2022-08-09
> > > **Response to the reviewer**
> > >
> > > Thank you for your thorough response.
> > >
> > > In the final version of the paper, we will make sure to clarify any confusion caused by the use of the terms “noise” and “channel.” You are entirely correct about substituting “channel” for “feature,” and we will try our best to correct this in the final version of the paper.
> > >
> > > Regarding your question about the z-score and how its threshold was determined, Mohit Agarwal and Raghupathy Sivakumar used the same threshold for z-score in their paper “Blink: A Fully Automated Unsupervised Algorithm for Eye-Blink Detection in EEG Signals,” where their task is similar to ours. We experimented (using only real data) with various z-score values and concluded that a value of 3 is the best possible value for our dataset as well. Outlier removal in health care domain signals is a standard process due to the noise from measurement tools. We will clarify these points in the final version of the paper.
> > >
> > > We should also emphasize that all of preprocessing steps were carried out on the real dataset and were based on the classification score on the real dataset. Furthermore, the preprocessed dataset was created and fixed before generating synthetic data and synthetic data evaluation. Although this is not the best practice for evaluating a classifier on real data, our preprocessing was agnostic to generative model evaluation and would not favor any generative model. We are confident that our results are a fair comparison of GroupGAN, baseline method, and SOTA methods because all methods benefit from the preprocessing steps. We hypothesize that our method will still outperform the others even if we do not remove the outliers.
> > >
> > > In response to your question about how forward feature selection was performed, we used sequential forward feature selection, as F.J. Ferri et al. discussed in their paper “Comparative study of techniques for large-scale feature selection”. In a nutshell, we performed a standard sequential forward feature selection by adding the locally best feature in the feature set regarding the classification score. When adding new features no longer improved the classification score, we stopped adding new features. In our case, no feature added to our top 5 would increase the classification score when trained on the real EEG dataset.
> > >
> > > And as for your final question, it is a valid concern for those interested in much higher dimensionality datasets that we will investigate and design experiments for in future works. Our results focus on helping the “middle-dimensionality” ( 2 < # features < ~10 ) that are ubiquitous in health care, for instance.
> > >
> > > Thank you once more for your time and consideration.

---

### Official Review · Reviewer_sRJW · 2022-07-11

**Rating:** 7
**Confidence:** 4
**Soundness:** 3 good
**Presentation:** 3 good
**Contribution:** 3 good

**Summary:**

In this paper, the authors propose Group GANs to generate Multivariate Time Series (MTS) data. MTS can be used for augmentation and privacy protection. To generate realistic MTS, the model should (1) generate realistic single-channel data (2) preserve the correlation between channels. So this paper uses a paired generator and discriminator to generate single-channel data and uses a Central Discriminator to force correlation preservation. They verify the results on synthetic and EEG datasets.

**Questions:**

Q1:

Eq (4) is the objective for “a given channel” i. What is the objective for all channels (for i generators + i discriminators + one central discriminator)?

Q2:

In eq (4), when updating channel i, does the Central Discriminator treat the generated samples in  other channels (j != i) as real ones? It seems the case because in line 100 you say “the goal is to estimate the conditional distribution”.

Q3:

What is the output of the central discriminator? I still don’t get it after reading Algorithm 1. The central discriminator takes in data of all the channels, but does it output a single score for all the channels combined or does it output a score for every channel (i scores in total)? And how do you update the central discriminator exactly (Describe the loss)?

**Limitations:**

More useful datasets and advanced model architectures can be used.

**Strengths And Weaknesses:**

## Strengths:
* The paper is well structured and nicely written. The authors provide a clear motivation for why generating MTS is useful and explain the key challenges.
* The authors tear down the problem of generating MTS into (1) generating realistic single-channel TS (2) preserving correlation. The authors propose Group GANs (many channel GANS + Central Discriminator) to address these problems. The method is straightforward and intuitive. I believe the proposed method will provide insight and inspire people working on related problems.
It’s good to see the authors submitted detailed supplementary materials.
* The authors use many metrics and methods (eg, Feature-based Analysis) to show how Diversity and Correlation are preserved. They also discussed the implementation details and trade-offs of their model, which can be valuable for the community.

## Weaknesses:
* I think the algorithm and training details are very important for understanding the paper. Maybe you can consider moving them (mainly Algorithm 1) from the Supplementary Materials to the body part.
* Some questions regarding the objectives remain. See Questions.
* The datasets are small, and they don’t look very practical. (As the authors state “our method could be extended to more practical use cases” in line 312).
* Whether the proposed model can scale very well remains a question. Performance limitations can be an issue. Another issue is that the authors only test MLP and LSTM, which is not a full guarantee that they can generalize to complex models like Transformers.

---

> ### Author Response · Authors · 2022-08-01
> **Answers to the questions**
>
> We will move the algorithm from supplementary materials to the main paper in the final release as we will have an extra page if the paper is accepted.
>
> In response to your first question, the objective for all channels is a `2C + 1` player game adding the terms with subscript `i` in equation (4). We will add a clarification sentence to the final release of the paper.
>
> In response to your second question, the central discriminator is fed two sets of stacked channels, the Real dataset, and the Synthetic dataset. Its goal is then to estimate the distribution of the Real dataset. The conditional distribution of a given synthetic channel depends on the other `fake` channels, which means that the central discriminator receives either all of the real channels or all of the fake channels. To avoid further misunderstandings, we will rewrite the aforementioned section if the paper is accepted.
>
> In response to your third question, the central discriminator is a binary classifier. As you mentioned, the output will be a single score for all channels combined. The central discriminator's weight update is a standard gradient ascent step based on the binary classification task's loss. We will include this information in the line mentioning the central discriminator in Algorithm 1.

---

> > ### Comment · Reviewer_sRJW · 2022-08-09
> > **Thanks for your response**
> >
> > Great! Thanks for your response. Hopefully you will make all the points clear in your revised version of the paper. My score remains the same.

---

### Official Review · Reviewer_NYDe · 2022-07-20

**Rating:** 7
**Confidence:** 3
**Soundness:** 3 good
**Presentation:** 2 fair
**Contribution:** 4 excellent

**Summary:**

This paper introduces GroupGAN a method for generating multivariate time series data that share a single source or cause. GroupGAN uses a separate generator and discriminator for generating each data channel and a central discriminator for forcing the model to keep the correlation between different channels the same as the real data. Synthetic data as well as EEG eye state dataset is used to evaluate GroupGAN compared to other models in this area.

**Questions:**

- On lines 151-154 you mention that for central discriminator MLP networks work better than LSTM networks because "if the central discriminator is too powerful, the results will be of lower quality, as it will strive to make the signals more correlated at the expense of realistic individual TS." can't this be controlled with the gamma weight?
- Is the "All-synthetic experiment" described on lines 243-255 the same as the "Train-on-fake, Test-on-real" experiment mentioned in table 3?
- In Figure 4, I wonder why GroupGAN does so much better than baseline on the All-Synthetic experiment while on the Augmentation experiment it is not doing as well? Is it because of a lower diversity of samples generated compared to the baseline?
- In Figure 4c, does augmenting more synthetic data reduce the performance of GroupGAN with respect to the baseline model, or does it improve it? It is not obvious from this figure.
- What is your hypothesis on why GroupGAN does not do as well on 2 channel datasets?

**Limitations:**

The limitations are discussed adequately.

**Strengths And Weaknesses:**

Strengths:

I believe the generation of multi-variate time-series data is a significant and very challenging problem. As mentioned by the authors it is urgently needed in many fields. The idea of using a central discriminator and separate generator discriminator pairs for each channel is very interesting and the authors show through experiments that this idea improves the data generation and downstream tasks.

Weaknesses:
- In the related works section it would be good to mention the weaknesses of the existing methods that the current model is trying to fix
- I think some additional comparisons would've strengthened the claim of the paper that GroupGAN works better than the baseline model (a single multi-channel GAN model for generating the data) for data generation (on lines 116 - 118), for example in tables 1 and 2 I expected to see a comparison with the baseline model.
- On line 155 the authors used LLD for the first time and then define them later it is better to define a term before using it.
- The paper can be improved by changes in language and writing style. There are typos and grammar problems that if fixed can improve the clarity of the paper. I bring some examples below:
- - on line 13 it should be "performs very well in downstream tasks"
- - on line 95 "channels" should be "channel"
- - on line 223 "EGG" should be "EEG"
-- on line 292 "compare" should be "comparison"

---

> ### Author Response · Authors · 2022-08-01
> **Answers to the questions**
>
> In response to your first question, the "gamma weight" influences the strength of the CD's effect in the loss calculation, whereas the architecture (LSTM vs. MLP) influences the ability to learn a "good" representation. Thus, a "reduced" loss strength of a "better" representation is not necessarily equivalent to an "increased" loss strength of a "worse" data representation.
>
> In answer to your second question, you are right, i.e., “All-Synthetic experiment” and “Train-on-Fake, Test-on-real” are the same concepts. However, the results provided in table 3 are for only two channels and compare the results of Group GAN with and without CD. We will clarify this point as well as other editorial suggestions in the final release of the paper if the paper is accepted.
>
> To answer your third question, It is common in synthetic data augmentation that "not so good" synthetic data can "appear" to help on average, but the variations can be huge, if not awful, sometimes. In the augmentation experiment, we aimed to demonstrate that one of the GroupGAN method's advantages is the "stability" of its results. As shown in figure 4 (b), in the case of 5 channels, the accuracy of the baseline method could be as high as 0.93 and as low as 0.6 among different synthetic data generations; however, the results of the Group GAN method show much more stability, and the accuracy of our method is much narrower, ranging from 0.77 to 0.94.
>
> Regarding your fourth question, increasing the augmentation ratio increases the accuracy and stability of both Group GAN and baseline methods, but not dramatically (less than 5 percent). However, in each distinct ratio, we conclude the same point we discussed in answer to your previous question: the Group GAN method is more stable than the baseline method.
>
> And regarding your last question, we suspect that two channels are too simple (tractable) a problem to require the Group GAN method and that even a simple baseline method could produce good enough synthetic data.

---

> > ### Comment · Reviewer_NYDe · 2022-08-10
> > **My score remains the same**
> >
> > Thank you for your response. My concerns have been addressed. I think this is a good paper and a score of 7 is appropriate. My score remains the same.

---

### Meta-Review · Area_Chair_9aXk · 2022-08-23

**Recommendation:** Accept
**Confidence:** Less certain

**Metareview:**

The authors propose GroupGAN which uses a separate generator and discriminator for generating each data channel and a central discriminator for accurately capturing the correlation structure between different channels. This is a borderline paper and there were extensive discussions among the reviewers about this paper. The reviewers all agreed that having two sets of discriminators is a good idea. However, the evaluation for this paper were only done with low-dimensional time series and the reviewers voiced concerns about the generalization of this method to higher dimensions. Some parts still need further clarification: for example, as Reviewer zeED pointed out, dimensionality reduction in this paper remains an extremely odd preprocessing step.

Overall, accepting this paper may start new discussions in the community. I hope the authors address the concerns raised by the reviewers in the camera-ready version of the paper.

**Award:**

No

---

### Decision · Program_Chairs · 2022-09-14

Accept